# Brain Metastases from Genito-Urinary Cancers in the Canton of Geneva (Switzerland): Study of Incidence, Management and Outcomes

**DOI:** 10.3390/cancers16203437

**Published:** 2024-10-10

**Authors:** Philippe Gonnet, Eliana Marinari, Vérane Achard, Robin Schaffar, Isabelle Neyroud-Caspar, Adrien May, Cristina Goga, Pierre-Yves Dietrich, Karl Schaller, Anna Patrikidou

**Affiliations:** 1Department of Internal Medicine, Hôpital de la Tour, 1217 Meyrin, Switzerland; 2Faculty of Medicine, University of Geneva, 1206 Geneva, Switzerland; eliana.marinari@unige.ch; 3Department of Radiation Oncology, HFR Fribourg, 1700 Fribourg, Switzerland; verane.achard@h-fr.ch; 4Geneva Cancer Registry, Global Health Institute, University of Geneva, 1206 Geneva, Switzerland; robin.schaffar@unige.ch (R.S.); isabelle.neyroud-caspar@unige.ch (I.N.-C.); 5Division of Neurosurgery, Department of Clinical Neuroscience, Geneva University Hospitals, 1205 Geneva, Switzerland; adrien.may@hcuge.ch (A.M.); karl.schaller@hcuge.ch (K.S.); 6Department of Neurosurgery, HFR Fribourg, 1700 Fribourg, Switzerland; cristina.goga@h-fr.ch; 7Division of Medical Oncology, Hirslanden Clinique des Grangettes, 1224 Chêne-Bougeries, Switzerland; pierre-yves.dietrich@hirslanden.ch; 8Department of Medical Oncology, Gustave Roussy Cancer Campus, 94805 Villejuif, France; anna.patrikidou@gustaveroussy.fr

**Keywords:** genito-urinary tumours, brain metastases, incidence, survival, prognostic factors, chemotherapy

## Abstract

**Simple Summary:**

Incidence of brain metastases from genito-urinary cancers has increased over the last years due to improved imaging techniques and better survival outcomes for patients even at an advanced metastatic stage thanks to therapeutic advances. However, no clear consensus exists on their management, and every case ought to be discussed in a multidisciplinary panel. We perform a single-centre retrospective study to report on incidence, patient demographics, clinicopathological characteristics, and treatment modalities. We also manage to identify predictive factors of outcome.

**Abstract:**

Background: Incidence of brain metastases is precisely unknown and there is no clear consensus on their management. We aimed to determine the incidence of brain metastases among patients with genito-urinary primaries, present patients’ characteristics and identify prognostic factors. Method: We identified 51 patients treated in Geneva University Hospitals between January 1992 and December 2019. We retrospectively correlated their overall survival with 23 variables. We repeated a multivariate analysis with significant variables. Results: Overall incidence of Brain Metastases (BMs) among Genito-Urinary (GU) patients is estimated to be 1.76% (range per primary GU tumour type: 0.00–6.65%). BMs originate from germ cell tumours in two cases (3.92%), from urothelial cell carcinoma in 15 cases (29.41%), from prostate cancer in 13 cases (25.49%), and from renal cell carcinoma in 21 cases (41.18%); there are no BMs from penile cancer in our cohort. The median age at BM diagnosis is 67 years old (range: 25–92). Most patients (54%) have a stage IV disease at initial diagnosis and 11 patients (22%) have BM at initial diagnosis. Only six patients (12%) are asymptomatic at BM diagnosis. The median Overall Survival (OS) from BM diagnosis is 3 months (range: 0–127). Five patients (10%) are long survivors (OS > 24 months). OS is significantly influenced by patient performance status and administration of systemic treatment. In the absence of meningeal carcinomatosis, OS is influenced by systemic treatment and stereotactic radiosurgery. We also apply the Graded Prognostic Assessment (GPA) score to our cohort and note significant differences between groups. Conclusion: Brain metastases from solid tumours is not a uniform disease, with a prognosis varying a lot among patients. The optimal management for patients with genito-urinary malignancies with brain metastases remain unclear and further research is needed.

## 1. Introduction

The global incidence of Central Nervous System (CNS) metastases is precisely unknown, but they are estimated to occur in 10–30% of all adult cancer patients, and their incidence is higher than that of primary CNS tumours [1,2].

In 2021, the American Society of Clinical Oncology (ASCO), the society of Neuro-oncology (SNO), and the American Society for Radiation Oncology (ASTRO) issued the first guidelines on the management of brain metastases (BMs) that address all possible interventions (systemic therapy, surgery, and radiation therapy) in a comprehensive fashion in one document. Patients with large BMs with mass effect are likely to benefit from surgery. When the primary cancer diagnosis is unknown, surgery might help assess the diagnosis and guide further management. When the systemic disease is not controlled and/or in the presence of several BMs, surgery is less likely to be beneficial. Local therapy (surgery, radiosurgery, and/or radiation therapy) should be given to patients with symptomatic brain metastases. Stereotactic Radiosurgery (SRS) alone is advised for patients with one-to-four unresected brain metastases (excluding small cell carcinoma). For one-to-two resected brain metastases, SRS is also beneficial. Whole Brain Radiation Therapy (WBRT), SRS, or their combination are conceivable options for patients with more than four unresected brain metastases or patients with more than two resected brain metastases and a Karnosky Performance Status (KPS) ≥ 70. When the brain metastases are asymptomatic, there is no benefit from radiotherapy if the KPS is ≤50% or if the KPS < 70 in the absence of systemic therapeutic options [3]. These guidelines were later completed by ASTRO in 2022, with recommendations focusing on radiotherapy. In the context of patients with an Eastern Cooperative Oncology Group (ECOG) score between 0 and 2, SRS is strongly advised for patients with up to four intact BMs and conditionally advised for patients with five-to-ten intact BMs. SRS is also recommended on resected BMs. When WBRT is administered, hippocampal avoidance is recommended [4]. These recommendations are presented in Figure 1.

Specific recommendations based on the primary cancer are available for a limited number of cancer types. A specific systemic treatment regimen can be offered to patients with BMs from melanoma, human epidermal growth receptor-2 positive breast cancer, EGFR-mutant, or ALK-rearranged non-small-cell-lung cancer [3]. For BMs originating from Renal Cell Carcinoma (RCC), cabozantinib is a recommended systemic option [5].

The development of prognostic indices, such as the Graded Prognostic Assessment (GPA) [6], has helped clarify the differential prognosis and outcome of BM patients according to disease-specific clinico-pathological characteristics, and, therefore, tailor therapeutic attitudes and treatment. However, the indicated management approaches and specific outcome for patients with Genito-Urinary (GU) malignancies and BM remain unclear, mostly owing to a lack of disease-specific data, with the exception of limited data on renal cancer BMs [7].

BMs are a very heterogeneous entity, depending on patient- and disease-related factors. The need for an in-depth knowledge of the natural history and specific characteristics of patients with BMs and their clinical outcome is required in order to establish the appropriate individual management strategy for the patient. Our study aims to obtain a comprehensive record and analysis of the incidence, characteristics, management and outcomes of patients with brain metastases from primary GU tumours in the canton of Geneva. Given the fact that almost the entirety of newly diagnosed BM lesions is addressed at the Cantonal University Hospital (HUG) and the approach implicates the involvement of the HUG neurosurgical and neuro-oncology team, analysis of the cases managed at HUG largely reflects the cantonal incidence.

## 2. Materials and Methods

### 2.1. Patients

The patients analysed in this project were selected from the electronic patient database of the HUG. All adult patients (male and female aged ≥ 18 years) with a primary genitourinary malignancy (prostate, bladder, renal, testicular, penile cancer) and a metastatic brain disease (parenchymal, meningeal, or both) managed at the Department of Oncology of HUG from January 1992 until December 2019 were included.

A coded search of the electronic patient records (EPR) was performed by the Data Analysis team of the Cancer Centre of the HUG on our behalf, using the ICDO-O classification codes of the different GU primary malignant pathologies (prostate cancer C81, urothelial cancer C65–C68, testicular cancer C62, renal cancer C64, penile cancer C80) in combination with the term “brain metastasis” or “brain tumour”. The identified cases were screened for eligibility with respect to the study criteria mentioned above.

Informed consent for data use was obtained from all alive patients. The HUG General Informed Consent Form was used. This study was part of a larger multicentre project on brain metastases of genitourinary tumours (BRAIN-GU project), for which approval from the local Research Committee of the Canton of Geneva was obtained (CCER 2017-01662). For deceased patients, we were allowed to use their data, unless it was mentioned in their EPR that they refused to share their data for medical research.

### 2.2. Data Collection

Patient demographics, clinicopathological characteristics at initial diagnosis and at BM diagnosis, BM number and localisation, presenting symptoms at BM diagnosis, BM treatment modalities, tumour assessment, and survival outcome data (Overall Survival [OS], Progression-Free Survival [PFS], Time-To-Brain Metastases [TTBM] from initial diagnosis and from diagnosis of metastatic GU cancer, brain PFS, extra-cranial PFS) were collected. Overall survival was defined as the time from the diagnosis of the brain metastatic disease until death (any cause) or until the patient was lost to follow-up. Progression-free survival was defined as the time from the diagnosis of the brain metastatic disease until the date of progression or until death (any cause).

Data extraction was done using the HUG EPR system. Dedicated Excel proforma databases were created for each GU type to allow for data collection. Data annotation and cleaning were performed at a second round of data verification. Patient data security was ensured by anonymisation, utilising a coded database named “Coded table”. Participants’ identifiable data (name and date of birth) were linked to a unique number (identification number) and stored in a password-protected file named “Identification table” in the Project Leader’s (AP) HUG PC.

To better assess the incidence of BM among GU patients in the canton of Geneva, we contacted the Geneva Cancer Registry to obtain anonymised incidence data.

### 2.3. Statistical Analyses

The incidence was initially calculated on the total number of GU oncology patients managed at the HUG Oncology Department during the same timeframe (1992–2019).

We assumed, however, that this incidence was overestimated, as many GU patients are treated outside the cantonal hospital, but most BM-GU patients are directed towards it for neurosurgical opinions and specialist management. The Geneva Canton Registry provided us with the incidence of GU tumours between 2005 and 2015. To calculate the incidence of BMs among GU patients, we only considered the number of BM-GU patients who had an initial diagnosis between 2005 and 2015.

Frequencies were calculated for noncontinuous variables. Median and ranges were calculated for continuous variables.

Patients without a survival event were censored at their last follow-up. We estimated survival curves using the Kaplan–Meier method for the overall population and per primary tumour type. Survival curves were compared using the log-rank test. To assess the prognostic effect of several characteristics on OS and PFS, we performed Univariate Analysis (UVA) using Cox proportional hazards model. The significance cut-off was set to *p* ≤ 0.05. We used the variables that were found to be statistically significant through UVA to perform a Multivariate Analysis (MVA).

We also assessed the application of the original Graded Prognostic Assessment (GPA) to this cohort of GU-BM patients [6].

The statistical analysis was performed in collaboration with a biostatistician from the Laboratory of Tumour Immunology, Translational Research Centre in Oncohæmatology, Faculty of Medicine, University of Geneva (EM). Statistical analysis was performed with the SPSS software v.25, IBM Corp, Armonk, NY, USA, 2017.

## 3. Results

### 3.1. Estimation of the Incidence of Brain Metastases from Genito-Urinary Primary Tumours

BM incidence amongst patients with GU malignancies in HUG between 1992 and 2019 and in the whole Geneva canton between 2005 and 2015 is presented in Table 1. The male:female ratio for renal and urothelial cancers was 2.3 (402 vs. 177 cases) and 3.6 (738 vs. 206 cases), respectively.

### 3.2. Patients and Disease Characteristics

From January 1992 to December 2019, 51 patients were assessed at the HUG regarding the management of their BM from a GU primary tumour. Among them, there were two patients with Germ Cell Tumours (GCT) (3.9%), 15 with Urothelial Cell Carcinoma (UCC) (29.4%), 13 with Prostate Cancer (PrCa) (25.5%), and 21 with Renal Cell Carcinoma (RCC) (41.2%). Due to the large amount of missing data, one PrCa patient treated outside HUG was excluded from the analysis. Out of 50 patients, 10 (20%) had BMs at the initial cancer diagnosis. Median age at diagnosis of primary GU tumour was 64 years, while it was 67 years for developing BM. Median TTBM was 35.5 months from initial diagnosis and 12 months from the metastatic GU disease. Patient characteristics are shown in Table 2. Five patients (10%) had an overall survival longer than 24 months and were considered long survivors. Their characteristics are presented in Table 3.

### 3.3. Outcomes

Median Follow-Up (FU) from BM diagnosis was 3 months (range 0–163) and only four patients (8%) were lost to FU (no data after initial diagnosis). Ten patients (20%) were alive at the end of FU, while 36 (72%) were deceased. Median extracranial PFS was 1 month (CI: 0.1–7.8) and the brain PFS was also 1 month (CI: 0.8–13.6). Median OS from BM diagnosis ranged from 2 months for UCC patients to 8.5 months for GCT patients, with a median OS of 3 months for the entire cohort (shown in Figure 2 and Table 4). Median OS did not significantly differ amongst GU primary tumours (*p* = 0.224, shown in Figure 3).

### 3.4. Univariate and Multivariate Analyses for Prognostic Factors

The following variables were significant for OS as a result of univariate analysis: age at BM diagnosis, Eastern Cooperative Oncology Group (ECOG) performance status or Karnofsky Performance Status (KPS) at BM diagnosis, presence of skin metastases, systemic treatment for the management of BM, Stereotactic Radiosurgery (SRS), surgery, Best Supportive Care (BSC), extracranial PFS, and brain PFS (Table 5). We assessed the prognostic ability of the number of BM lesions for patients without neoplastic meningitis and the presence of a single BM significantly improved OS (*p* = 0.016, HR = 0.305).

As ECOG PS and Karnofsky Performance Status subgroups repartition was identical, we only present the data for ECOG PS. Through multivariate analysis, OS was improved with an ECOG PS 0–1 (HR = 0.190; 95% CI: 0.048–0.760; *p* = 0.019) and the administration of a systemic treatment (HR = 0.335; 95% CI: 0.115–0.977; *p* = 0.045). For the subcohort of patients without neoplastic meningitis, SRS also improved OS (HR: 0.095; 95% CI: 0.011–0.845; *p* = 0.035) (Table 6).

### 3.5. Implementation of the Original Graded Prognostic Assessment Score

Most patients (23 out of 50) belonged to the poor prognostic group (score: 0–1) and had a median survival of 2 months. There was no patient in the best prognostic group (score: 3.5–4). The differences among groups are illustrated in Figure 4 (*p* < 0.000).

## 4. Discussion

### 4.1. The HUG/Geneva Canton Experience

Our study provides a detailed description of patients presenting BM from primary GU tumours in the canton of Geneva. This is the first comprehensive report of incidence, management, and outcomes with respect to a Swiss population, and, to our knowledge, it is the first collective report on BM from genito-urinary malignancies in the body of international literature.

BM incidence in our study (Table 1) was comparable to that reported in the literature, with ranges in the following intervals: 0.5–2.6% for GCT [8,9], 0.25–7% for UCC [10,11,12,13], 0.07–2% for PrCa [14,15,16,17,18], and 1.48–11% for RCC [19,20,21,22].

Unsurprisingly, most patients presented with an advanced stage at initial diagnosis, indicating aggressive tumours, also corroborated by the relatively short median OS from initial diagnosis (28.5 months). Incidence data for asymptomatic BM are scarce in the literature and are mostly available for lung cancer [23]. Of interest, 12% of our overall cohort and 20% of our BM at initial diagnosis subcohort presented with asymptomatic BM, and their survival was longer compared to that of the patients with symptomatic BM (5 months vs. 3 months for the overall cohort), consistently with previous reports on other malignancies [24].

Our long survivor group did not feature an age difference compared to the overall cohort, but it exhibited an overwhelming good performance status (80% had KPS of 90–100 vs. 26% for the overall cohort) and a unique BM that principally occurred as a late relapse (median TTBM of 50 months).

Our study confirms the strong prognostic role of patient performance status with respect to patients with brain metastases from GU tumours, in line with prognostic indices for all major tumour types [25,26]. Equally unsurprisingly, the number of brain lesions influenced OS.

Systemic treatment was the second statistically significant factor influencing overall survival in our cohort. This finding supports the role of tailored systemic treatment in this setting, both in the context of multimodality treatment for BM but also for the control of extracranial diseases, which very frequently dictate the prognosis in these patients.

SRS was significant when the analysis was restricted to the patients without meningeal carcinomatosis, despite the smaller patient number. This finding supports, for a careful selection of patients, the use of local modalities: patients with good performance status, limited number of BM, and absence of meningeal diseases.

A major limitation of our study is the small size and heterogeneity of our cohort, given the different primary tumour types. Indeed, the small number of patients in the individual subcohorts did not allow for meaningful analyses with respect to the primary GU tumours, with such analyses having the potential to highlight the intricacies of each tumour type.

### 4.2. Current State-of-the Art on Brain Metastases from Genito-Urinary Malignancies

Limited data exist in the body of international literature on BM management from GU primaries, mostly in the form of small case series. The presence of BMs leads to the classification of germ cell tumours as having a poor prognosis [27]. Extrapulmonary visceral metastases and a mediastinal primary present a higher risk of BM occurrence [28]. Upfront, BMs account for 2–3% of advanced GCTs and for 10–15% of poor-prognosis GCTs. Their 3-year OS is 45–50%. BMs at relapse have an even poorer prognosis, and cure is only possible by multimodality treatment in individual cases (3-year OS 25–30%, 5-year OS 2–5%). Prognostic factors include multiple BMs and the presence of liver or bone metastases. In patients with BMs at initial diagnosis, presence of primary mediastinal nonseminoma cancers is also a negative prognostic factor. In patients with BMs at relapse, elevation of alpha-fetoprotein above 100 ng/mL or human chorionic gonadotropine above 5000 U/L are additional adverse prognostic factors. In this second group, a high dose of chemotherapy and multimodality treatment seem to improve survival probabilities [29,30]. A brain MRI is recommended at diagnosis in case of suspicious symptoms, widespread metastases (especially if supradiaphragmatic), and very high levels of HCG. Brain relapses occur frequently within the first 1–2 years (≈40%), and survival is worse if relapse presents within 1 year. BMs are predominant (≈55%) in cases of radiographic progression only (no tumour marker rise) [31]. The optimal treatment remains unclear, but multimodality is pivotal, especially at relapse [30]. Chemotherapy is the standard of care and ought to be started before local treatment [28,29,30]. The role of radiotherapy is poorly defined and it entails late neurotoxicity [32]. SRS is indicated for unresectable or very small isolated residual BMs post-chemotherapy (or sandwiched between chemotherapy sessions), whilst WBRT should be avoided. Surgery is a relatively uncommon scenario, reserved for accessible, solitary, or limited residual masses with a good extracranial response post-chemotherapy and normalised markers. Ongoing clinical trials have been evaluating other approaches, including immunotherapy, with moderate efficacy so far [33,34].

Limited data exist in the literature for BMs from metastatic urothelial tumours, encompassing mostly case reports or limited series, given the rarity of BMs as part of this tumour, and universally confirm their poor prognosis [35,36]. In most cases, these are symptomatic at diagnosis. One of the largest of such series (169 patients) has reported a prognostic nomogram for bladder cancer patients with BMs, indicating that chemotherapy offers the maximal survival benefit for these patients [37]. Brain metastasectomy remains controversial and is reserved for unique lesions in the context of controlled extracranial diseases in patients with a good performance status [38]. In 2023, a case series with patients treated with enfortumab vedotin showed promising results [39].

Parenchymal brain metastases and leptomeningeal disease are rare in association with prostate cancer and represent a late manifestation with a poor prognosis, frequently due to neuroendocrine clonal differentiation of the disease. Nevertheless, skull bone metastases in patients suffering from prostate cancer can frequently cause a reaction/invasion of the adjacent dura by continuity (pachymeningeal metastases) [40,41].

Overall survival for patients presenting BMs from RCC ranges from 6 months to 10 months [19]. Recommendations for the management of advanced RCC have evolved in the recent years. Intermediate results from the CaboPoint trial encourage the use of cabozantinib in the context of advanced RCC after progression on checkpoint inhibitors [42]. Its efficacy in targeting brain metastases has been demonstrated in a recent study with an intracranial response rate of 55% and a median overall survival of 16 months [43]. Ongoing studies are evaluating the safety and efficacy of nivolumab with ipilimumab and cabonzantinib in patients with untreated RCC BMs [44].

## 5. Conclusions

It is increasingly recognised that brain metastases from solid tumours are not a uniform disease, nor are they subject to the same management and outcome principles as extracranial metastatic disease. The optimal management approaches and specific outcomes for patients with genito-urinary malignancies and brain metastases remain unclear, mostly owing to a lack of disease-specific data. Although their incidence has traditionally been underestimated, BMs affect up to one third of adults with cancer and represent an important public health burden that is 10 times more common than malignant primary brain tumours [26]. As oncologists face a growing number of patients with BMs from GU cancer primaries, the need for an in-depth knowledge of the natural history and specific characteristics of patients with BMs and their clinical outcomes is required in order to establish the appropriate individual management strategy for the patient. Our project was developed in this direction and offers the first insights into the Swiss reality. However, given the rarity of these patients, large multicentre international projects involving pooled data analyses are needed to correctly assess management approaches, determine consensus, and optimise outcomes.

## Figures and Tables

**Figure 1 cancers-16-03437-f001:**
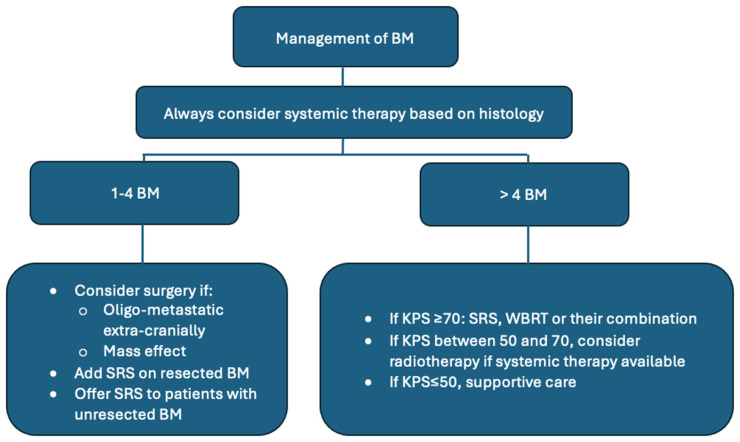
Management of patients with brain metastases.

**Figure 2 cancers-16-03437-f002:**
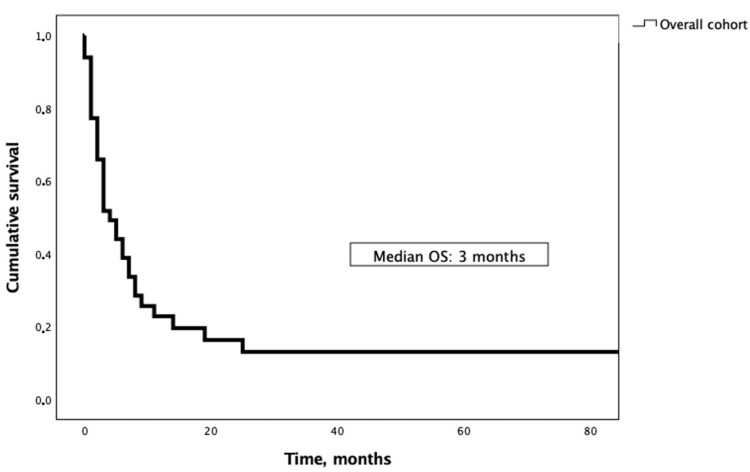
Overall survival from brain metastasis diagnosis.

**Figure 3 cancers-16-03437-f003:**
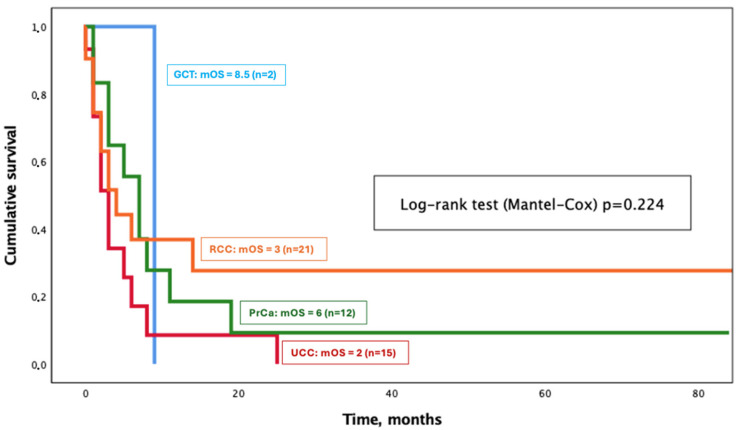
Overall survival from brain metastasis diagnosis per genito-urinary tumour type.

**Figure 4 cancers-16-03437-f004:**
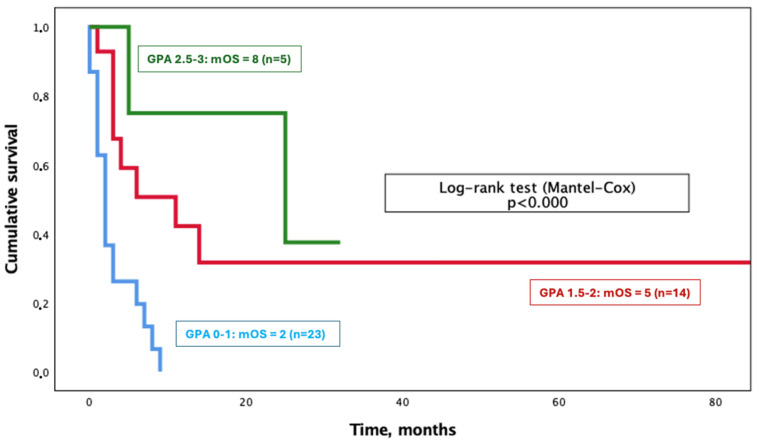
Overall survival from brain metastasis diagnosis according to the Graded Prognostic Assessment score.

**Table 1 cancers-16-03437-t001:** Incidence of BM among patients with a primary genito-urinary tumour.

GU Type	Geneva University Hospital Patients (1992–2019)			Geneva Canton Patients (2005–2015)		
	Patients with GU Tumours	Patients with BM	BM Incidence (%)	Patients with GU Tumours	Patients with BM	BM Incidence (%)
GCT	134	2	1.49	207	0	0.00
UCC	1094	15	1.37	944	13	1.37
PrCa	1340	13	0.97	3347	9	0.27
RCC	316	21	6.65	579	17	2.94
Penile Ca	18	0	0.00	36	0	0.00
All GU	2902	51	1.76	5113	39	0.76

BM: Brain Metastases, Ca: Cancer, GCT: Germ Cell Tumours, GU: Genito-Urinary, HUG: Geneva University Hospitals, PrCa: Prostate Cancer, RCC: Renal Cell Carcinoma, UCC: Urothelial Cell Carcinoma.

**Table 2 cancers-16-03437-t002:** Overall cohort characteristics.

Characteristics	Overall	GCT	UCC	PrCa	RCC
	*n* = 50, *n* (%)	*n* = 2, *n* (%)	*n* = 15, *n* (%)	*n* = 12, *n* (%)	*n* = 21, *n* (%)
Age at diagnosis, years					
Median	64	51	71	62.5	62
Range	25–92	36–66	42–89	50–92	25–83
Gender					
Female	8 (16)		3 (20)		5 (23.8)
Male	42 (84)	2 (100)	12 (80)	12 (100)	16 (76.2)
Histology					
		NSGCT: 2 (100)	TCC: 15 (100)	AdenoCa: 10 (83)	Clear cell: 16 (76.2)
				AdenoCa with small cell component: 1 (8.3)	Xp11 translocation: 1 (4.76)
					Sarcomatoid differentiation: 1 (4.76)
MD				Unkown: 1 (8.3%)	Unknown: 3 (14.3)
Stage at initial diagnosis					
I	2 (4)				2 (9.5)
II	5 (10)		3 (20)		2 (9.5)
III	12 (24)	1 (50)	3 (20)	2 (16.7)	6 (28.6)
IV	27 (54)	1 (50)	7 (46.7)	10 (83.3)	9 (42.8)
MD	4 (8)		2 (13.3)		2
Presence of BMs at initial diagnosis	11 (22)	1 (50)	5 (33.3)	2 (16.7)	3 (14.3)
Age at BM diagnosis					
Median	67	51.5	73	65.5	67
Range	25–92	36–67	47–90	61–92	25–88
ECOG PS at BM diagnosis					
0–2	39 (78)	2 (100)	10 (66.7)	9 (75)	18 (85.7)
3–4	11 (22)		5 (33.3)	3 (25)	3 (14.3)
KPS at BM diagnosis					
90–100	13 (26)	1 (50)	4 (26.7)	2 (16.7)	6 (28.6)
70–80	25 (50)	1 (50)	5 (33.3)	7 (58.3)	12 (57.1)
50–60	4 (8)		1 (6.7)	1 (8.3)	2 (9.5)
≤40	8 (16)		5 (33.3)	2 (16.7)	1 (4.8)
BM localisation					
Only parenchymal lesions	34 (68)	2 (100)	11 (73.3)	3 (25)	18 (85.7)
Supratentorial	22 (44)	2 (100)	7 (46.7)	2 (16.7)	11 (52.4)
Infratentorial	5 (10)		1 (6.7)	1 (8.3)	3 (14.3)
Both	15 (30)		7 (46.7)	2 (16.7)	6 (28.6)
Only meningeal carcinomatosis	8 (16)			7 (58.3)	1 (4.8)
Both	8 (16)		4 (26.7)	2 (16.7)	2 (9.5)
Number of parenchymal metastases					
1	17 (34)	1 (50)	4 (26.7)	2 (16.7)	10 (47.6)
2–3	6 (12)	1 (50)	1 (6.7)	1 (8.3)	3 (14.3)
>3	19 (38)		10 (66.7)	2 (16.7)	7 (33.3)
Site of metastasis at BM diagnosis					
Lung metastasis	31 (62)	1 (50)	7 (46.7)	3 (25)	20 (95.2)
Bone metastasis	28 (56)		5 (33.3)	12 (100)	11 (52.4)
Liver metastasis	13 (26)		2 (13.3)	4 (33.3)	7 (33.3)
Skin metastasis	2 (4)				2 (9.5)
Lymph nodes	37 (74)	2 (100)	10 (66.7)	8 (66.7)	17 (81)
Other	13 (26)		Gastro-intestinal: 1 (6.7)	Splenic and adrenal: 1 (83.3)	Adrenal gland: 2 (9.5), controlateral kidney: 2 (9.5), intramuscular: 4 (19), parotid gland: 1 (4.8), pleurae: 1 (4.8), peritoneum: 2 (9.5), spleen: 1 (4.8), thyroid: 1 (4.8)
BM presenting symptoms					
None	6 (12)	1 (50)			5 (23.8)
Headache	7 (14)		1 (6.7)	2 (16.7)	4 (19)
Motor deficit	12 (24)	1 (50)	4 (26.7)	3 (25)	4 (19)
Sensory deficit	7 (14)	1 (50)	1 (6.7)	4 (33.3)	1 (4.8)
Seizure	8 (16)	1 (50)	2 (13.3)	2 (16.7)	3 (14.3)
Confusion	11 (22)		6 (40)	2 (16.7)	3 (14.3)
Nausea, vomiting	1 (2)			1 (8.3)	
Aphasia or speech troubles	8 (16)		5 (33.3)	2 (16.7)	1 (4.8)
Visual disturbances	6 (12)		3 (20)	3 (25)	
Ataxia	7 (14)		3 (20)	1 (8.3)	3 (14.3)
Neuropsychological	10 (20)		7 (46.7)		3 (14.3)
Other	3 (6)		1 (6.7)	1 (8.3)	1 (4.8)

AdenoCa: Adenocarcinoma, ECOG PS: Eastern Cooperative Oncology Group Performance Status, GCT: Germ Cell Tumours, GU: Genito-Urinary, KPS: Karnofsky Performance Status, MD: Missing Data, NSGCT: Non-Seminoma Germ Cell Tumour, PrCa: Prostate Cancer, RCC: Renal Cell Carcinoma, TCC: Transitional Cell Carcinoma, UCC: Urothelial Cell Carcinoma.

**Table 3 cancers-16-03437-t003:** Long survivors’ characteristics.

Characteristics	Long Survivors *n* = 5, *n* (%)
Gender	Male: 5 (100)
Histology	UCC—TCC: 1 (20)
	Prostate—adenocarcinoma: 1 (20)
	RCC—clear cell: 3 (60)
PSA at initial diagnosis (prostate)	MD: 1
ISUP Gleason score (prostate)	Intermediate unfavourable grade: 1 (20)
Fuhrmann Grade (RCC)	
3	1 (20)
4	1 (20)
MD	1 (20)
Stage at initial diagnosis	
I	1 (20)
II	1 (20)
III	1 (20)
IV	2 (40)
BM at initial diagnosis	1 (20)
Age at BM diagnosis	
Median	66
Range	36–75
ECOG PS at BM diagnosis	
0–1	4 (80)
2	1 (20)
KPS at BM diagnosis	
90–100	4 (80)
70–80	1 (20)
GPA at initial diagnosis	
1	5 (100)
BM localisation	
Only parenchymal lesions	5 (100)
Localisation of parenchymal metastases	
Supratentorial	4 (80)
Infratentorial	1 (20)
Both	
Number of parenchymal metastases	
Only 1	5 (100)
Sites of metastasis at BM diagnosis	
Lung metastasis	3 (60)
Bone metastasis	1 (20)
Liver metastasis	0 (0)
Skin metastasis	0 (0)
Other	3 (60)

BM: Brain Metastases, ECOG PS: Eastern Oncology Collaborative Group Performance Status, ISUP: International Society of Urological Pathology, KPS: Karnofsky Performance Status, MD: Missing Data, PSA: Prostate Specific Antigen, RCC: Renal Cell Carcinoma, TCC: Transitional Cell Carcinoma, UCC: Urothelial Cell Carcinoma.

**Table 4 cancers-16-03437-t004:** Outcomes.

Outcomes	Overall	GCT	UCC	PrCa	RCC
	*n* = 50, *n* (%)	*n* = 2, *n* (%)	*n* = 15, *n* (%)	*n* = 12, *n* (%)	*n* = 21, *n* (%)
FU length (months)					
Median	3	8.5	2	45.5	3
Range	0–163	8.5	0–63	1–163	0–127
Last known FU state					
Alive and disease free	1 (2)	1 (50)			
Alive without PD	2 (4)				2 (9.5)
Alive with brain PD	5 (10)		2 (13.3)		3 (14.3)
Alive with extracranial PD	2 (4)			1 (8.3)	1 (4.8)
Lost to FU	4 (8)			1 (8.3)	3 (14.3)
Deceased—no further information	1 (2)				1 (4.8)
Deceased due to brain PD	15 (30)		9 (60)	1 (8.3)	5 (23.8)
Deceased due to extracranial PD	12 (24)		1 (6.7)	7 (58.3)	4 (19)
Deceased due to treatment complication	2 (4)		1 (6.7)	1 (8.3)	
Deceased due to another pathology	6 (12)	1 (50)	2 (13.3)	1 (8.3)	2 (9.5)
Last known ECOG PS					
0–2	9 (18)	1 (50)	1 (6.7)		7 (33.3)
3–4	5 (10)		1 (6.7)	2 (16.7)	2 (9.5)
5	36 (72)	1 (50)	13 (86.7)	10 (83.3)	12 (57.1)
Last known KPS					
90–100	5 (10)	1 (50)	1 (6.7)		3 (14.3)
70–80	4 (8)				4 (19)
50–60	0 (0)				
10–40	5 (10)		1 (6.7)	2 (16.7)	2 (9.5)
0	36 (72)	1 (50)	13 (86.7)	10 (83.3)	12 (57.1)
Brain radiological progression					
Yes	15 (30)	1 (50)	5 (33.3)	2 (16.7)	7 (33.3)
MD	27 (54)		9 (60)	7 (58.3)	11 (52.4)
Clinical progression	42 (84)	1 (50)	14 (93.3)	11 (91.7)	16 (76.2)
PFS from BM treatment					
Extracranial PFS from BM treatment					
Median	1	7	0	1.5	0
Range	0–90	5–9	0–23	0–7	0–90
95% CI	0.1–7.8	−18.4–32.4	−0.9–5.6	1–3.4	−3.8–16
MD	2 (4)	0 (0)	0 (0)	0 (0)	2 (9.5)
Brain PFS from BM treatment					
Median	1	5	0	5.5	0
Range	0–127	5	0–20	0–84	0–127
95% CI	0.8–13.6	NA	1.2–4.4	−29.3	−28.4
MD	2 (4)	0 (0)	0 (0)	0 (0)	2 (9.5)
OS from initial diagnosis					
Median	28.5	12.5	19	57	28
Range	0–273	9–16	0–110	4–243	1–273
95% CI	34–68.8	−32–57	14–50.5	31.5–122.3	22.7–85.6
MD	0 (0)				
OS from BM diagnosis					
Median	3	8.5	2	6	3
Range	0–127	8–9	0–25	1–84	0–127
95% CI	3.4–15.7	2.1–14.9	0.7–7.5	−1.9–27.4	−1.1–24.4
For patients with BM at initial diagnosis, OS from BM diagnosis					
Median	3	8	3	3	3
Range	0–11	8	0–8	3	1–11
95% CI	1.9–7.1		−0.2–7.8		−8.1–18.1
Disease-specific OS from initial diagnosis					
Median	23.5		9	63	37
Range	0–184		0–110	4–163	1–184
95% CI	28.9–70.7		−2.6–45.2	27.3–110.5	13.5–111.1
NA	22 (44)	2 (100)	5 (33.3)	3 (25)	12 (57.1)
Disease-specific OS from BM diagnosis					
Median	2		2	5	2
Range	0–19		0–8	1–19	0–14
95% CI	2.2–5.8		0.9–4.9	1.9–10.7	−0.4–6.2
NA	22 (44)	2 (100)	5 (33.3)	3 (25)	12 (57.1)

BM: Brain Metastases, CI: Confidence Interval, ECOG PS: Eastern Collaborative Oncology Group Performance Status, FU: Follow-Up, GCT: Germ Cell Tumours, KPS: Karnofsky Performance Status, MD: Missing Data, NA: Non-Applicable, OS: Overall Survival, PD: Progressive Disease, PFS: Progression-Free Survival, PrCa: Prostate Cancer, RCC: Renal Cell Carcinoma, UCC: Urothelial Cell Carcinoma.

**Table 5 cancers-16-03437-t005:** Univariate analysis.

	Effect on OS			
	*p*	HR	95%CI	
			Lower	Upper
Gender (reference: female)	0.464	0.700	0.269	1.820
Tumour type (reference: GCT)				
UCC	0.195	3.844	0.500	30.198
PrCa	0.433	2.282	0.290	17.959
RCC	0.491	2.058	0.264	16.060
Age at BM diagnosis	**0.001**	1.053	1.021	1.085
BM at initial diagnosis vs. relapse	0.986	0.993	0.430	2.293
Number of relapses at BM diagnosis	0.296	1.139	0.892	1.453
Time to BM from initial diagnosis	0.351	0.997	0.989	1.004
Time to BM from mGU diagnosis	0.155	1.009	0.996	1.022
ECOG PS at BM diagnosis	**0.000**	2.249	1.573	3.214
ECOG PS at BM diagnosis				
(reference: 3–4)				
2	**0.001**	0.135	0.041	0.441
0–1	**0.000**	0.082	0.027	0.248
KPS at BM diagnosis (continuous)	**0.000**	0.946	0.926	0.967
KPS at BM diagnosis				
(reference: <60)				
60–70	**0.001**	0.135	0.041	0.441
80–100	**0.000**	0.082	0.027	0.248
Site of metastasis at BM diagnosis				
Lung metastasis	0.972	0.988	0.511	1.910
Bone metastasis	0.079	1.844	0.932	3.648
Liver metastasis	0.091	1.973	0.897	4.341
Skin metastasis	**0.008**	8.096	1.732	37.835
Other	0.859	1.075	0.484	2.388
When no NM—Number of BM				
(reference: >3)				
2–3	0.653	0.744	0.205	2.698
1	**0.016**	0.305	0.116	0.804
Presence of meningeal carcinomatosis	0.394	1.351	0.677	2.697
BM localisation				
(reference: only parenchymal metastases)				
Only NM	0.532	1.312	0.559	3.079
NM and BM	0.470	1.401	0.561	3.497
BM localisation				
(reference: supratentorial)				
Infratentorial	0.637	0.701	0.161	3.059
Supratentorial and infratentorial	0.196	1.688	0.763	3.735
Systemic treatment for BM	**0.020**	0.488	0.228	0.881
WBRT	0.880	0.946	0.463	1.935
SRS	**0.038**	0.466	0.226	0.960
Surgery	**0.035**	0.446	0.210	0.945
Type of resection (reference: complete)				
Subtotal ≥ 90%	0.848	1.168	0.237	5.750
Partial: 50–90%	0.783	1.352	0.158	11.556
BSC (reference: no BSC)	**0.000**	9.270	3.724	23.075
Radiologic progression (brain)	0.403	1.660	0.506	5.445
Clinical progression	0.124	26.012	0.411	1647.3
Extracranial PFS from BM treatment	**0.043**	0.864	0.751	0.995
Brain PFS from BM treatment	**0.010**	0.878	0.795	0.969

BM: Brain Metastases, BSC: Best Supportive Care, CI: Confidence Interval, ECOG PS: Eastern Collaborative Oncology Group Performance Status, GCT: Germ Cell Tumour, GU: Genito-Urinary, HR Hazard Ratio, IGCCCG: International Germ Cell Cancer Collaborative Group, ISUP: International Society of Urothelial Pathology, KPS: Karnofsky Performance Status, NM: Neoplastic Meningitis, OS: Overall Survival, PFS: Progression-Free Survival, PrCa: Prostate Cancer, PSA: Prostate Specific Antigen, RCC: Renal Cell Carcinoma, SRS: Stereotactic Radiosurgery, UCC: Urothelial Cell Carcinoma, WBRT: Whole-Brain Radiation Therapy. Statistically significant values (*p* < 0.05) are in bold.

**Table 6 cancers-16-03437-t006:** Multivariate analysis for overall survival.

	Overall (*n* = 50)				Without Meningeal Carcinomatosis (*n* = 34)			
	*p*	HR	95%CI		*p*	HR	95%CI	
			Lower	Upper			Lower	Upper
Age at BM diagnosis	0.224	1.024	0.986	1.063	0.417	1.029	0.961	1.102
ECOG PS at BM diagnosis								
(reference: 3–4)								
2	0.448	0.564	0.128	2.477	0.538	2.262	0.168	30.4
0–1	**0.019**	0.190	0.048	0.760	0.127	0.188	0.022	1.607
Presence of skin metastasis	0.384	2.303	0.353	15.040	0.276	5.739	0.247	133.337
BSC (reference: no BSC)	0.742	1.237	0.349	4.382	0.396	0.396	0.046	3.372
Systemic treatment for BM	**0.045**	0.335	0.115	0.977	**0.033**	0.137	0.022	0.85
SRS	0.396	0.641	0.230	1.788	**0.035**	0.095	0.011	0.845
Surgery	0.342	0.624	0.236	1.649	0.394	0.402	0.049	3.274
Brain PFS from BM treatment	0.116	0.917	0.823	1.022	0.095	0.932	0.858	1.012
Extracranial PFS from BM treatment	0.377	0.936	0.808	1.084	0.381	0.91	0.738	1.123
Number of BM (reference: >3)								
2–3					0.205	7.465	0.334	167.032
1					0.75	1.274	0.287	5.665

BM: Brain Metastases, BSC: Best Supportive Care, CI: Confidence Interval, ECOG PS: Eastern Collaborative Oncology Group Performance Status, GU: Genito-Urinary, HR: Hazard Ratio, KPS: Karnofsky Performance Status, OS: Overall Survival, PFS: Progression-Free Survival, SRS: Stereotactic Radiosurgery. Statistically significant values (*p* < 0.05) are in bold.

## Data Availability

The data that support the findings of this study are not publicly available due to their containing information that could compromise the privacy of research participants.

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
