# Peer review of "Brain Metastases from Genito-Urinary Cancers in the Canton of Geneva (Switzerland): Study of Incidence, Management and Outcomes"

_cancers, 2024, doi:10.3390/cancers16203437_

Round 1

Reviewer 1 Report

Comments and Suggestions for Authors

The authors here provide one review article to summarize the findings from primary 101 genitourinary malignancy patients with brain metastases. Overall this is one informative paper for us obtaining the relative data. I may have some suggestions to the authors.

1, how about the influence of different types of cancer treatment for the brain metastasis patients? The authors should consider dividing the cohorts of patients with their therapeutic approaches.

2, why do the different cancers cause different brain metastasis rates? The authors should have some explanation on this especially for urothelial cell carcinoma, prostate cancer and renal cell carcinoma. It seems these 3 cancer types possess much longer BM rates compared to the others.

3, In the notes of the tables, some terms are in italic form, some are not. Please reconcile the format of them.

4, For the table 2, the contents are too complicated to follow up, Please reformat the table to make it concise. If there is too much information, please separate the table into several sub tables.

5, For the figure 3, there is one zigzag red line under the mOS, please remove it.

6, some short name of brain metastases is BM, some is BMs, please make them the same.

Comments on the Quality of English Language

The English is fine for publication.

Author Response

Dear Reviewer,

Thank you very much for taking the time to review this manuscript. Please find the detailed responses below and the corresponding corrections in tracked changes in the re-submitted manuscript

Comments 1: how about the influence of different types of cancer treatment for the brain metastasis patients? The authors should consider dividing the cohorts of patients with their therapeutic approaches.

Response 1: Thank you for your suggestion. We acknowledge that we do not formally present the data based on the therapeutic approaches. Most of our patients underwent a combination of treatment modalities. As treatment combinations are numerous, we did not divide their presentation accordingly to the treatment combinations. However, in our univariate and multivariate analysis (Tables 5 and 6), patients are divided based on their therapeutic approaches, showing that systemic treatment, stereotactic radiosurgery and surgery significantly impact on overall survival. We believe that the small size of our cohort would not allow us to further divide patients into further subgroups based on treatment modalities.

Comments 2: why do the different cancers cause different brain metastasis rates? The authors should have some explanation on this especially for urothelial cell carcinoma, prostate cancer and renal cell carcinoma. It seems these 3 cancer types possess much longer BM rates compared to the others.

Response 2: Thank you for your comment. It’s true that we did not discuss why brain metastases rates vary depending on the primary tumour. To our knowledge, there are no definite answers. Metastatization to the brain depends on several factors: the preferential way a primary tumour metastizes (blood, lymph node), compatibility with the brain microenvironment and efficacy of treatment. We did not believe that adding information about the factors favoring metastatization to the brain would be useful in the frame of our manuscript.

Comments 3: In the notes of the tables, some terms are in italic form, some are not. Please reconcile the format of them.

Response 3: Thank you for your comment. We updated the format so that all the abbreviations are in italics, and their explanation are upright.

Comments 4: For the table 2, the contents are too complicated to follow up, Please reformat the table to make it concise. If there is too much information, please separate the table into several sub tables. 

Response 4: Thank you for the comment. We added lines to separate the table and make it more readable.  

Comments 5: For the figure 3, there is one zigzag red line under the mOS, please remove it.

Response 5: Thank you for your comment. We removed these red lines. We also removed these red lines from figure 2.

Comments 6: some short name of brain metastases is BM, some is BMs, please make them the same.

Response 6: Thank you for your comment. We changed all abbreviations to BM.

Reviewer 2 Report

Comments and Suggestions for Authors

I read with great interest the article by Gonnet et al. titled Brain metastases from genito-urinary cancers in the Canton of Geneva (Switzerland): Study of incidence, management, and outcomes,” which was recently submitted to the Cancers (Basel) journal.

The authors are to be commended for providing useful knowledge for the incidence, management, and outcomes of brain metastases (BM) from genito-urinary cancers. They present their experience through a retrospective analysis between January 1992  and December 2019. Notably, to the author’s knowledge, it is the first study on a Swiss population and the first collective report on BMs from genito-urinary malignancies in the international literature.

They also describe the latest advancements in the field of treatment of various urological malignancies with BM.

Although it has a small sample size, the present article is sufficiently detailed. The tables presented are pretty descriptive. Overall, the manuscript is well-written, clear, and concise in the most significant part. The syntax is appropriate, and the use of English is acceptable.

o   Abstract  

significative differences  >  better change it to significant 

o   Figures 

Fig 2 and Fig 3 have the spelling corrections underlined. 

Please omit them and erase the red lines from the figures (e.g., mOS)

Author Response

Dear Reviewer,

Thank you very much for taking the time to review this manuscript. Please find the detailed responses below and the corresponding corrections in tracked changes in the re-submitted manuscript

Comments 1:  Abstract  : significative differences  >  better change it to significant 

Response 1: Thank you for your comment. We changed the manuscript accordingly.

Comments 2: Fig 2 and Fig 3 have the spelling corrections underlined. Please omit them and erase the red lines from the figures (e.g., mOS)

Response 2: Thank you for your comment. We removed the red lines from the figures.

Reviewer 3 Report

Comments and Suggestions for Authors

The manuscript entitled "Brain metastases from genito-urinary cancers in the Canton of Geneva (Switzerland): Study of incidence, management and outcomes" is well written in understandable English. It covers an important theme of brain metastases incidence in case of genito-urinary cancers. The authors retrospectively analyzed the data over a considerable period of time and their conclusions seems to be strong.

In general, the masnucript can be accepted for publication, however I have some minor remarks which should be corrected:

1. line 49 "and their incidence is more important than that of primary CNS tumours [1, 2]." Why it is more important?

2. Line 65 first mention of KPS, provide deabbreviation here

3. Lines 51-72. Please, provide graphic scheme illustrating optimal ways for cancer treatment in different cases of BM presence.

4. Line 11: "An informed consent for data use was obtained from all alive patients". And what about dead patients?

5. Table 1 "PenileCa" format text in one line

Author Response

Dear Reviewer,

Thank you very much for taking the time to review this manuscript. Please find the detailed responses below and the corresponding corrections in tracked changes in the re-submitted manuscript.

Comments 1: line 49 "and their incidence is more important than that of primary CNS tumours [1, 2]." Why it is more important?

Response 1: Thank you for your comment. We meant that the incidence of brain metastases is higher than the incidence of central nervous system primaries. We changed the formulation.

Comments 2: Line 65 first mention of KPS, provide deabbreviation here

Response 2: Thank you for your comment. We changed the manuscript accordingly.

Comments 3: Lines 51-72. Please, provide graphic scheme illustrating optimal ways for cancer treatment in different cases of BM presence.

Response 3: Thank you for your comment. We provided a scheme for illustrating the management of brain metastases.

Comments 4:  "An informed consent for data use was obtained from all alive patients". And what about dead patients?

Response 4: Thank you for your comment. We added in our manuscript: “For deceased patients, we were allowed to use their data, unless it was mentioned in their EPR that they refused to share their data for medical research.”

Comments 5: Table 1 "PenileCa" format text in one line

Response 5: Thank you for your comment. We modified the table accordingly.